# Influence of Playing Standard on Upper- and Lower-Body Strength, Power, and Velocity Characteristics of Elite Rugby League Players

**DOI:** 10.3390/jfmk4020022

**Published:** 2019-04-17

**Authors:** John F. T. Fernandes, Matthew Daniels, Liam Myler, Craig Twist

**Affiliations:** 1Sport, Health and Well-being, Hartpury University, Hartpury GL19 3BE, UK; 2St Helens Rugby League Club, St Helens WA9 3AL, UK; 3Widnes Vikings Rugby League Club, Widnes WA8 7DZ, UK; 4Department of Sport and Exercise Sciences, University of Chester, Chester CH1 4BJ, UK

**Keywords:** physical qualities, profiling, youth, adult, muscle function

## Abstract

Background: To compare load–velocity and load–power relationships among first grade (*n* = 26, age 22.9 ± 4.3 years), academy (*n* = 23, age 17.1 ± 1.0 years), and scholarship (*n* = 16, age 15.4 ± 0.5 years) Super League rugby league players. Methods: Participants completed assessments of maximal upper- and lower-body strength (1RM) and peak velocity and power at 20, 40, 60, and 80 kg during bench press and squat exercises, in a randomised order. Results: Bench press and squat 1RM were highest for first grade players compared with other standards (effect size (ES) = −0.43 to −3.18). Peak velocities during bench and squat were greater in the higher playing standards (ES = −0.39 to −3.72 range), except for the squat at 20 and 40 kg. Peak power was higher in the better playing standards for all loads and exercises. For all three groups, velocity was correlated to optimal bench press power (*r* = 0.514 to 0.766), but only 1RM was related to optimal power (*r* = 0.635) in the scholarship players. Only squat 1RM in the academy was related to optimal squat power (*r* = 0.505). Conclusions: Peak velocity and power are key physical qualities to be developed that enable progression from junior elite rugby league to first grade level. Resistance training should emphasise both maximal strength and velocity components, in order to optimise upper- and lower-body power in professional rugby league players.

## 1. Introduction

Rugby league is a contact sport that requires players to possess a range of physical qualities for success [1]. Of these qualities, muscular strength and power might assist in the effective execution of several skills that determine performance or player selection. For example, upper-body strength and power have strong relationships (*r* = 0.72 and 0.70, respectively) with tackling ability [2], while upper- and lower-body strength and power are able to differentiate between playing standards in rugby league players [3,4]. Upper-body power was only different between state and national standard rugby league players at higher external loads of 70 and 80 kg [5], suggesting that power exerted against high external loads is a key discriminator of success in rugby league players. Baker and Newton [6] also reported that upper- and lower-body strength and power characteristics were able to better distinguish between rugby league playing standards than other measures of acceleration, maximal speed, and agility. 

Baker and Nance [7] reported strong correlations between upper-body strength and power (*r* = 0.89) and lower-body strength and power (*r* = 0.81) in professional rugby league players. However, the relationship between strength and power might well be influenced by playing standard, with lower standard players presenting better associations (*r* = 0.85) than national standard (*r* = 0.58) players [3]. This observation suggests that the training emphasis is likely to be different between players of different standards, with important implications for those designing resistance training programmes for the long-term development of rugby players. Regarding the contribution of barbell velocity to power output, Fernandes and colleagues [8] reported that velocity was not related to bench press power in young resistance trained males. During the squat exercise, velocity was also moderately correlated (*r* = 0.653) with power in these males [8]. Interestingly, in stronger individuals, velocity appears to underpin adaptation to the lower-body power movements [9]. A study in well-trained rugby league players that determines the contribution of both strength and velocity to power during upper- and lower-body resistance exercises would enable a closer examination of the interplay between these neuromuscular characteristics. 

While recent studies have examined differences in physical qualities of senior, academy, and youth rugby league players [1], measures of maximal strength, load–power, and load–velocity between rugby players of different training ages have not been provided before. In rugby union athletes, Hansen and colleagues [10] noted that elite athletes (~26 years) produced higher power during the 40 kg jump squat exercise than their junior counterparts (~19 years) from the same team. However, the single load selected by Hansen et al. [10] means that it is unknown if the differences in power exist at lower and higher loading conditions.

The primary aim of this study was to provide a detailed comparison of the load–velocity and load–power relationship among rugby league players of different playing standards within the same club. A secondary aim is to establish the contribution of strength and velocity to upper and lower body power in rugby league players.

## 2. Materials and Methods

### 2.1. Participants

Twenty-six first grade (age 22.9 ± 4.3 years), 23 academy (age 17.1 ± 1.0 years), and 16 scholarship (age 15.4 ± 0.5 years) rugby league players competing in the Super League were recruited for the study. These groups comprised the entire playing squad of each team, with only injured players exempt from taking part in the study. All participants regularly performed bench press and squats as part of their resistance training programme. Participants completed informed consent and a pre-test health questionnaire for the study, which was approved by the Ethics Committee (072/16/LM/SES on 20/07/16) at the University of Chester. Parental consent was attained for those under 18 years.

### 2.2. Design

Participants completed measurements of body mass and body composition followed by the maximal bench press and squat exercise. Thereafter, participants completed three repetitions of bench press and squat at four absolute loads (20, 40, 60, and 80 kg). Only four of the scholarship players could perform the 80 kg bench press, meaning only their data from 20–60 kg were analysed. We opted to use absolute loading conditions, rather than relative, as this better reflects match demands. That is, players are required to express velocity and power against absolute loads, irrespective of their individual strength. Such an approach has been adopted previously [3,4,5,6,7]. The testing battery was performed at the end of an eight-week pre-season training phase focusing on maximal strength and power development. The testing battery had been performed previously with the players, meaning they were habituated to the procedures. 

### 2.3. Procedures

#### 2.3.1. Physical Measurements

Body mass was determined using calibrated digital scales (Seca 813; Seca, Hamburg, Germany) and body composition was estimated from the sum of skinfold thickness (mm) from bicep, triceps, pectoral, subscapular, iliac crest, supraspinale, abdominal, front thigh, and medial calf. Skinfold thickness was taken twice (Harpenden, Holtain, Crymych, Dyfed, UK) at each site and if the difference between measurements were <5%, the mean score was used for analysis. Where the difference was ≥5%, a third measurement was taken, and the median value was used for analysis.

#### 2.3.2. Strength Testing

Participants’ maximum strength on bench press exercise was assessed directly using a standardised 1RM protocol [11]. For safety reasons, 1RM during squat exercise was predicted from a 3RM as detailed by Baker and Newton [6]. This method estimates maximal strength on the basis that a 3RM is 93% of the 1RM (i.e., (3RM load/93)*100) [12]. Previous data indicate that this method provides a reliable assessment of maximal strength (intraclass correlation coefficients and coefficient of variation (CV) of 0.91 and 3.6%, respectively) [2]. Relative upper- and lower-body strength was calculated by dividing 1RM by body mass.

#### 2.3.3. Assessment of Peak Velocity and Power

Peak velocity and power were determined during the bench press and squat exercise at four absolute loads: 20, 40, 60, and 80 kg. Loads were applied in a randomised order with measurements of peak velocity and power being recorded using the FitroDyne rotary encoder (Fitronic, Bratislava, Slovakia) attached via nylon cord directly under the end of a barbell. The FitroDyne provides reliable measures of peak velocity (CV = 2.1% to 8.8%) and power (CV = 2.2% to 8.5%) at a range of external loads [13]. 

For the bench press exercise, participants held the barbell with a prone grip and lowered it to their chest before pushing maximally. During the squat exercise, participants descended with the barbell across their shoulder until their hips were below the knee joint and then ascended as rapidly as possible until their knees were at full extension. Three repetitions of each exercise were performed at each load with rest intervals of two minutes between repetitions. The average of three repetitions was selected for analysis.

### 2.4. Statistical Analyses

Differences in dependent variables were examined using Bayesian analysis, which employed the effect size (ES) with associated 90% confidence intervals (CI) [14]. This method is a form of ‘calibrated’ Bayes inference with a dispersed uniform prior. Moreover, this approach allowed for a more practical and meaningful explanation of the data that are deemed more useful to the coach and athlete when determining the magnitude of the differences. Thresholds for the magnitude of the observed difference for each variable were determined as the within-participant standard deviation in that variable × 0.2, 0.6, and 1.2 for a small, moderate, and large effect, respectively [15]. Threshold probabilities for a meaningful effect based on the 90% CI were as follows: <0.5% most unlikely, 0.5%–5% very unlikely, 5%–25% unlikely, 25%–75% possibly, 75%–95% likely, 95%–99.5% very likely, >99.5% most likely. Effects with CI across a likely small positive or negative difference were classified as unclear [14]. All calculations were completed using predesigned spreadsheets (www.sportsci.org). Data are presented as ES ± CI. Readers should be aware of the recent debate regarding the use of this approach, particularly concerning the error rates (see Sainani [16] and www.sporrtsci.org). Partial correlation coefficients were calculated to provide an estimation of the contribution of maximal velocity (at 20 kg) and 1RM to power at the load that optimised power (40 and 80 kg for bench press and squat, respectively). For all partial correlations, the variables not being analysed were controlled for (e.g., the relationship between velocity and power, controlling for 1RM). Alpha was set at 0.05. These data were analysed in SPSS (Version 24, IBM SPSS Inc., Chicago, IL, USA).

## 3. Results

### 3.1. Physical Characteristics

There were small to large differences in body mass between groups with mean values higher in the first grade group compared with other groups (Table 1). Sum of skinfolds was moderately lower in the first grade players compared with academy players, but no differences were observed for any other comparison. Moderate to large differences in absolute (kg) and relative to body mass (kg·bm^−1^) bench press and squat strength reflected better performance in higher playing standards.

### 3.2. Peak Velocity

There were large differences in peak velocity for bench press at all loads, with first grade players outperforming both academy and scholarship players, while those of academy players were also greater than those of scholarship players. Conversely, differences in peak velocity during squat exercise between first grade and academy players was small at 20 and 40 kg, despite large differences at 60 and 80 kg. Similarly, there were small differences in squat peak velocity at 20 kg between first grade and scholarship players, but large differences at 40, 60, and 80 kg. Moreover, the comparison between first grade and scholarship players reflected widening group differences with an increasing load. An analysis of academy and scholarship players’ data revealed small to moderate differences in squat peak velocity. All data are shown in Figure 1 and Table 2.

### 3.3. Peak Power

Large differences in bench press peak power were observed between first grade academy and scholarship players at all external loads. The small to moderate differences in bench press peak velocity between academy and scholarship players at 20 and 40 kg were accompanied by large differences at 60 kg. For all comparisons, the magnitude of the differences between groups was not related to the external load. For squat peak power, differences between all comparisons and at all external loads were large, except for 20 kg between academy and scholarship players, where differences were moderate. The magnitude of the differences between groups only appeared to differ across external loads in the first grade versus scholarship comparison (i.e., greater differences with increasing external load). All data are shown in Figure 2 and Table 2.

### 3.4. Partial Correlations

When controlling for bench press velocity, 1RM was only correlated with optimal power in the scholarship players (*r* = 0.635, *p* < 0.05, Table 3). Correlations for 20 kg velocity to optimal power were moderate to strong in all groups (*r* = 0.514 to 0.788, *p* < 0.05). For the squat exercise, only 1RM was correlated to optimal power (*r* = 0.505, *p* < 0.05) in the academy group.

## 4. Discussion

This is the first study to provide a detailed analysis of the load–velocity and load–power relationships between rugby league players of different playing standards. These findings indicate that peak velocity and power are key descriptors of playing standard in rugby league players and thus provide a training progression for academy and scholarship players.

First grade players had a greater body mass than both academy and scholarship players, with academy values being higher than scholarship values. This is similar to previous reports of an increased body mass with playing standard [10,17,18,19] and likely reflects differences in maturation [20,21]. The lower body mass alongside higher sum of skinfolds in the academy players compared with their first grade counterparts would indicate a higher amount of fat mass and lower fat-free mass. Furthermore, the sum of skinfolds was not different for any other comparison. In support, Till and colleagues [22] observed comparable skinfold values across 15 to 20 year old rugby league players. The fact that body mass increased with playing standard, but the sum of skinfolds did not for the first cf. scholarship and academy players compared with scholarship players suggests a greater fat-free mass in the higher playing standards. A greater fat-free mass in the higher playing standards might be attributable to the players’ resistance training exposure. For example, the scholarship and academy players’ exposure to resistance training took place recently (<2 years), while the first grade players had been regularly exposed to resistance training for longer (>7 years). Importantly, a lower skinfold thickness score is associated with enhanced skill related performance (e.g., sprinting, change of direction [23]), but also supports the importance of a higher mass coupled with faster sprint speeds in senior player to optimise momentum into the collision [1].

As expected, the first grade players had greater absolute and relative upper- and lower-body strength than academy and scholarship players. Scholarship players were also weaker, in both absolute and relative terms, than academy players for both exercises. Comparable differences in upper- [3,4,21,22] and lower-body [6,10,21,22] strength, between playing standards, have been reported previously. Like body mass, these strength differences might be explained by maturity and training age of the participants. A greater fat-free mass in senior players, indicated by a higher body mass and lower skinfold thickness, might also contribute to the higher force production in senior players [24,25]. Together, these data reaffirm that upper- and lower-body maximum strength are key descriptors of playing standard between rugby league athletes.

Excluding the squat at 20 kg for all groups and 40 kg between first grade and academy players, peak velocity typically demonstrated moderate to large differences between groups. To our knowledge, no study has examined upper-body pushing velocity across different playing standards. As such, we report, for the first time, that bench press velocity is able to distinguish between rugby league players of different training ages. The fact that lower-body velocity is able to differentiate between playing standard is in support of a previous investigation in rugby union [10], but contrasts reports in Australian rules players, where there were no differences observed between higher and lower standards [26]. Notably, our study expands on previous work in that velocity was determined at a range of external loads rather than unloaded [26] or single-loaded [10] conditions. Rugby league players are expected to produce efforts against a range of loaded conditions, for example, sprinting and tackling. These differences in velocity might be explained by the greater strength with higher playing standards, and thus the absolute loadings accounting for a lower percentage of 1RM in the higher playing standards. Moreover, morphological (e.g., greater amount of type 2 fibres, pennation angle) and neurological (e.g., decreased antagonist coactivation, motor unit synchronisation) differences [6,24,25,27] might provide a more mechanistic explanation of the differences observed in the current study. Practically, strength and conditioning coaches should aim to improve upper- and lower-body velocity at a range of external loads as players progress from lower to higher playing standards. 

Peak power, similar to strength, reflected playing standard for all exercises and loads. That is, the first grade expressed higher peak powers than the academy and scholarship players, with academy values being greater than scholarship values. These data support previous observations in both upper- [3,4,21] and lower-body power [4,6,10,21,26]. Given that power is the product of force (strength) and velocity, these differences between playing standards are likely owing to the differences in strength and velocity between groups. Therefore, the higher power with playing standard can be explained by greater lean mass; maturation; training age; and, plausibly, morphological and neurological differences [6,20,21,22,23,27]. Collectively, these data suggest that the enhancement of power, alongside other physical qualities [1], is a pathway for progression in rugby league players.

For the bench press, strength was moderately correlated to optimal power in the scholarship players, but not first grade or academy players. The notion that the relationship between strength and power is decreased with playing standard has been observed previously [4,21]. These data suggest that once players are relatively strong enough (i.e., a 1RM of >1.3 kg·bm^−1^, as for the first grade and academy players), then other physical attributes must be focused upon. Indeed, the relationship between velocity and optimal power was moderate to strong for the first grade, academy, and scholarship players (*r* = 0.514, 0.546, and 0.788, respectively). Only one study [8] has generated comparable data, whereby velocity was strongly correlated to optimal power during the bench press in young resistance trained males. This suggests that high peak powers are achieved through greater velocity in better playing standards. During the squat exercise, only the academy players’ strength was correlated to optimal power. This reaffirms previous data [7], but contrasts observations of no relationship between lower-body strength and power [21]. The reason for the weak associations between lower-body strength and optimal power in the current study is unclear. Other factors, such as rate of force development [28], might be of more importance in these populations and future studies should determine this empirically.

## 5. Conclusions

Irrespective of the external load, both load–velocity and load–power relationships during the bench press and squat exercise reflect playing standard in professional rugby league players. Regardless, when increasing squat peak power, academy players should aim to increase their maximal lower-body strength. Early training focus for upper- and lower-body training should emphasise the development of maximal force generation. As players progress towards senior rugby, resistance training should also include developing a player’s ability to exert maximal barbell velocity and power during a bench press and back squat against a range of external loads.

## Figures and Tables

**Figure 1 jfmk-04-00022-f001:**
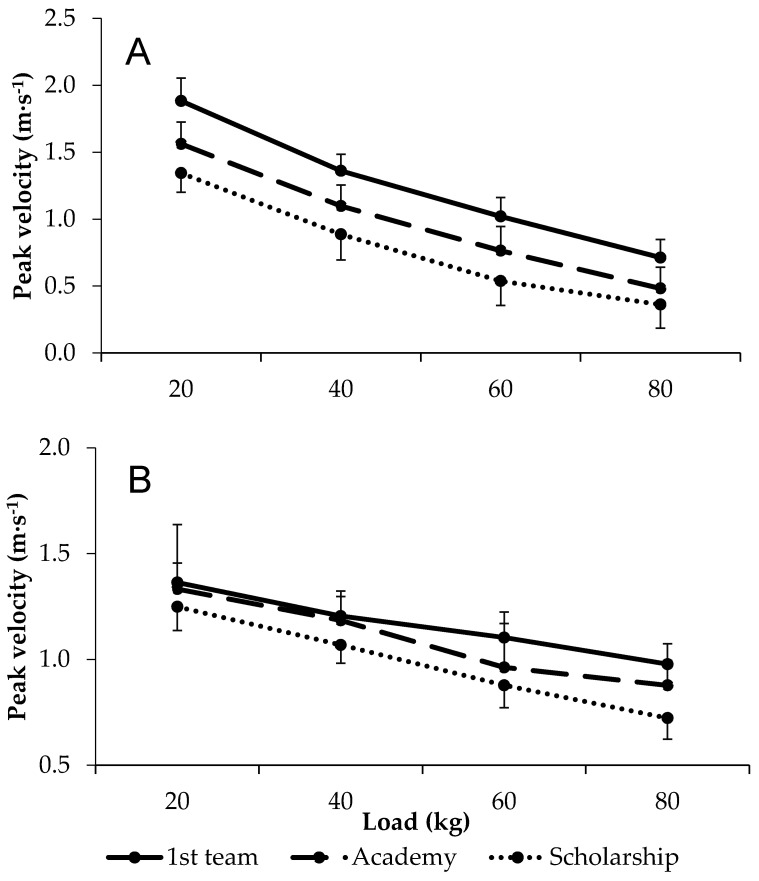
Load–velocity relationships in first team, academy, and scholarship players during the (**A**) bench press and (**B**) squat exercise.

**Figure 2 jfmk-04-00022-f002:**
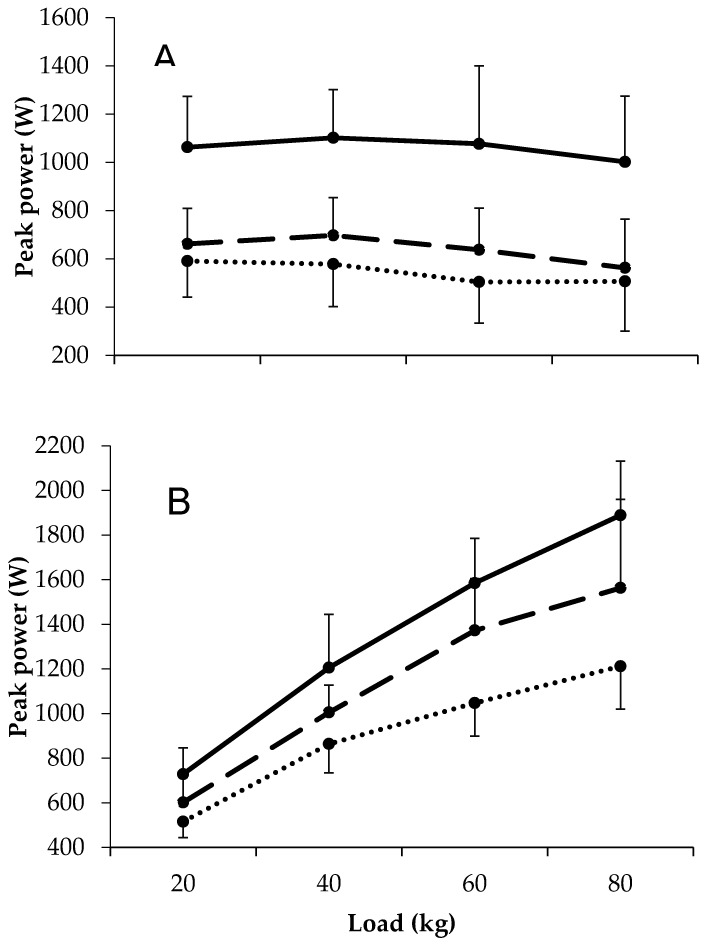
Load–power relationships in first team, academy, and scholarship players during the (**A**) bench press and (**B**) squat exercise.

**Table 1 jfmk-04-00022-t001:** Biometric characteristics (mean ± SD) of the first team, academy, and scholarship players. Qualitative descriptor, effect size ±90% confidence intervals are noted in the effect size column.

				Effect Size		
	First Grade (*n* = 26)	U’19s (*n* = 23)	U’16s (*n* = 16)	First vs. Academy	First vs. Scholarship	Academy vs. Scholarship
Mass (kg)	94.6 ± 9.5	85.9 ± 10.4	79.7 ± 10.8	−0.89 ± 0.49	−1.52 ± 0.57	−0.58 ± 0.55
				Very likely	Most likely	Likely
Sum of skinfolds (mm)	81.0 ± 14.7	90.7 ± 23.9	88.2 ± 29.3	0.65 ± 0.64	0.48 ± 0.90	−0.10 ± 0.60
				Likely	Unclear	Unclear
Bench press 1RM (kg)	135.2 ± 16.2	111.5 ± 14.3	82.2 ± 12.6	−1.42 ± 0.44	−3.18 ± 0.46	−1.98 ± 0.50
				Most likely	Most likely	Most likely
Relative bench press 1RM (kg·bm^−1^)	1.43 ± 0.14	1.30 ± 0.15	1.03 ± 0.12	−0.87 ± 0.50	−2.76 ± 0.47	−1.71 ± 0.46
				Very likely	Most likely	Most likely
Squat 1RM (kg)	183.3 ± 20.6	174.3 ± 27.0	140.0 ± 22.2	−0.43 ± 0.53	−2.04 ± 0.56	−1.23 ± 0.48
				Possibly	Most likely	Most likely
Relative squat 1RM (kg·bm^−1^)	1.94 ± 0.22	2.04 ± 0.26	1.78 ± 0.32	−0.78 ± 0.92	−0.71 ± 0.70	−0.94 ± 0.61
				Likely	Likely	Very likely

**Table 2 jfmk-04-00022-t002:** Qualitative interpretation and effect size (ES) ± confidence interval (CI) for the interpretation of dependent variables during the bench press and squat exercise.

		20 kg		40 kg		60 kg		80 kg	
		Velocity	Power	Velocity	Power	Velocity	Power	Velocity	Power
First grade vs. Academy	Bench press	−1.83 ± 0.46	−1.85 ± 0.40	−2.06 ± 0.54	−1.97 ± 0.42	−1.76 ± 0.54	−1.32 ± 0.37	−1.66 ± 0.55	−1.56 ± 0.44
		Most likely	Most likely	Most likely	Most likely	Most likely	Most likely	Most likely	Most likely
	Squat	−0.11 ± 0.36	−1.04 ± 0.45	−0.17 ± 0.46	−0.81 ± 0.37	−1.14 ± 0.67	−1.03 ± 0.51	−1.01 ± 0.52	−1.31 ± 0.65
		Unclear	Most likely	Unclear	Most likely	Very likely	Most likely	Very likely	Most likely
First grade vs. scholarship	Bench press	−3.07 ± 0.47	−2.17 ± 0.43	−3.72 ± 0.72	−2.55 ± 0.49	−3.32 ± 0.65	−1.72 ± 0.39		
		Most likely	Most likely	Most likely	Most likely	Most likely	Most likely		
	Squat	−0.41 ± 0.36	−1.75 ± 0.41	−1.13 ± 0.44	−1.39 ± 0.39	−1.82 ± 0.48	−2.61 ± 0.44	−3.11 ± 0.54	−2.71 ± 0.46
		Unclear	Most likely	Most likely	Most likely	Most likely	Most likely	Most likely	Most likely
Academy vs. Scholarship	Bench press	−1.29 ± 0.49	−0.47 ± 0.54	−1.31 ± 0.61	−0.73 ± 0.57	−1.21 ± 0.55	−0.75 ± 0.54		
		Most likely	Likely	Most likely	Likely	Most likely	Very likely		
	Squat	−0.66 ± 0.51	−0.75 ± 0.43	−1.00 ± 0.46	−1.11 ± 0.55	−0.39 ± 0.40	−1.35 ± 0.43	−1.28 ± 0.49	−0.85 ± 0.39
		Likely	Very likely	Most likely	Most likely	Likely	Most likely	Most likely	Most likely

**Table 3 jfmk-04-00022-t003:** Partial correlations for velocity (controlling for 1RM) and 1RM (controlling for velocity) with optimal power.

	Bench Press		Squat	
	1RM	Velocity	1RM	Velocity
First team	0.310	0.514 *	0.365	0.117
Academy	0.310	0.546 *	0.505 *	0.256
Scholarship	0.635 *	0.788 *	0.332	0.484

* denotes significant correlation (*p* < 0.05).

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
