# Peer review of "Influence of Playing Standard on Upper- and Lower-Body Strength, Power, and Velocity Characteristics of Elite Rugby League Players"

_jfmk, 2019, doi:10.3390/jfmk4020022_

Reviewer 1 Report

Review of:

Influence of playing standard on upper- and lower-body strength, power and velocity characteristics of elite rugby players: A Case study

The authors should be commended for what is a nice simple study with a good sample size. The paper is well written and flows well. A couple of very minor suggestions before publication:

Line 92-95; could the authors provide a reliability value for the 1RM and 3RM protocols as well as the prediction equation for back squat 1RM.

Line 100-101; Since the authors state that the device is reliable, the reliability value of the FitroDyne device should be stated.

Author Response

We thank the three reviewers for their overwhelmingly positive appraisal of our paper and for the opportunity to resubmit for their reconsideration. We feel that the manuscript is now much improved and hope it meets with the reviewers’ approval. The comments have been amended as suggested (detail in red within the manuscript) within the paper.

 Reviewer 1

 The authors should be commended for what is a nice simple study with a good sample size. The paper is well written and flows well. A couple of very minor suggestions before publication:

 Line 92-95; could the authors provide a reliability value for the 1RM and 3RM protocols as well as the prediction equation for back squat 1RM.

 -       A reliability value has been provided on lines 94-96; ‘Previous data indicates that this method provides a reliable assessment of maximal strength (intraclass correlation coefficients and coefficient of variation (CV) of 0.91 and 3.6%, respectively) [2].’ Regarding the prediction equation, the sentence on lines 93 and 94 now reads ‘This method estimates maximal strength on the basis that a 3RM is 93% of the 1RM (i.e. (3RM load / 93)*100).’

 Line 100-101; Since the authors state that the device is reliable, the reliability value of the FitroDyne device should be stated.

 -       We have updated this sentence to include the reliability values; ‘The FitroDyne provides reliable measures of peak velocity (CV = 2.1 to 8.8%) and power (CV = 2.2 to 8.5%) at a range of external loads [12].’

Reviewer 2 Report

General Comments

This is an excellent study that has generated some very interesting data on strength and power characteristics in different levels of rugby league players.  The manuscript is well written and easy to follow. I have some specific comments below regarding the wording in places, but these are minor.  It would be worthwhile incorporating the research by James and colleagues (2018) (https://www.ncbi.nlm.nih.gov/pubmed/29281133) that investigated the impact of strength level on changes in peak velocity.   The study should be of high interest to practitioners and scientists working in the rugby codes.

Specific Comments

Title  I would recommend removing “A case study”.  Given the number of players tested, I think this is more than a case study.

Line 46 Suggest rewording the sentence “Only one study has determined…” as many studies have investigated this relationship (although not using the approach you are alluding to and particularly in rugby league players). 

Line 49  Add the r-value for correlation between velocity and power in your previous study?

Line 66-67  ± symbols missing here?

Line 70  Also note here that parental/guardian consent was provided for those under the age of 16?

Line 76  Include a brief rationale (here or somewhere in the discussion) as to why you chose to use absolute loads for the testing.

Line 84  Add space between “…) and body…”

p.3  Include the reliability of the 1RM test in your club/laboratory. Or if not available, note the reliability of these measures in similar populations.  For the peak velocity/power, it would be worth mentioning the specific reliability results from previous work e.g. range of ICC’s, CV.s.

Line 125 “Physical” rather than “Biometric”?

p.5, Figure 1 Check formatting of this figure – “B” Y-axis should start at zero?

Line 189-90  Use “compared to” rather than “cf”

Line 198  Change to “Scholarship”

Line 224 Change to “scholarship”

Line 235  Make it clear here you are talking about bench press relative 1RM strength.

References Need to fix repeat numbering of reference list.

Reference #11  Update to 4th Edition of Essentials Text (Haff and Triplett).

Author Response

We thank the three reviewers for their overwhelmingly positive appraisal of our paper and for the opportunity to resubmit for their reconsideration. We feel that the manuscript is now much improved and hope it meets with the reviewers’ approval. The comments have been amended as suggested (detail in red within the manuscript) within the paper.

 Title I would recommend removing “A case study”.  Given the number of players tested, I think this is more than a case study.

-       Case study has been removed

Line 46 Suggest rewording the sentence “Only one study has determined…” as many studies have investigated this relationship (although not using the approach you are alluding to and particularly in rugby league players).

-       This sentence has been amended to ‘Regarding the contribution of barbell velocity to power output, Fernandes and colleagues [8] reported that velocity was not related to bench press power in young resistance trained males.’

Line 49  Add the r-value for correlation between velocity and power in your previous study?

-       The r value has now been included; ‘During squat exercise, velocity was also moderately correlated (r = 0.653) to power in these males [8].’

Line 66-67  ± symbols missing here?

-       This has been amended.

Line 70  Also note here that parental/guardian consent was provided for those under the age of 16?

-       Parental consent was attained for the those under 18. This has been included in the manuscript; ‘Participants completed informed consent and a pre-test health questionnaire for the study, which was approved by the Ethics Committee of the host institution. Parental consent was attained for those under 18 years.’

Line 76  Include a brief rationale (here or somewhere in the discussion) as to why you chose to use absolute loads for the testing.

-       A brief rationale has been included in the methods section; ‘We opted to use absolute loading conditions, rather than relative, as this better reflects match demands. That is, players are required to express velocity and power against absolute loads, irrespective of their individual strength. Such an approach has been adopted previously [3-7].’

Line 84  Add space between “…) and body…”

-       This has been amended

 p.3  Include the reliability of the 1RM test in your club/laboratory. Or if not available, note the reliability of these measures in similar populations.  For the peak velocity/power, it would be worth mentioning the specific reliability results from previous work e.g. range of ICC’s, CV.s.

-       the data from previous work has now been included ‘Previous data indicates that this method provides a reliable assessment of maximal strength (intraclass correlation coefficients and coefficient of variation (CV) of 0.91 and 3.6%, respectively) [2].’

Line 125 “Physical” rather than “Biometric”?

-       This has now been changed to physical (amended in the methods too).

 p.5, Figure 1 Check formatting of this figure – “B” Y-axis should start at zero?

-       The size of the graph has been amended to improve the clarity

 Line 189-90  Use “compared to” rather than “cf”

-       This has been amended

 Line 198  Change to “Scholarship”

-       This has been amended

 Line 224 Change to “scholarship”

-       This has been amended

 Line 235  Make it clear here you are talking about bench press relative 1RM strength.

-       This has been amended to ‘These data suggest that once players are relatively strong enough (i.e. a 1RM of > 1.3 kg·bm-1) then other physical attributes must be focused upon.’

 References Need to fix repeat numbering of reference list.

-       The references have been edited correctly

 Reference #11  Update to 4th Edition of Essentials Text (Haff and Triplett).

-       This reference has been edited

Reviewer 3 Report

The research group investigated an important aspect of physical prerequisites and playing level. However, in my view, the goal of a case study, better pilot observation, is to find better hypotheses. So, the authors made their conclusion to early based on correlation instead of causational modeling.

If we take into account that the professional players have significant more training history it is no surprise that they have shown a higher maximum strength and peak power levels compared to academy and scholarships levels. However, I have some, conceptional methodological and causational clues for the authors.

I would recommend starting to frame the article with a rugby performance model and trying to integrate these findings in this model.

Are there athletes with weak strength levels, which also show high rugby performance in your data or in practice like in football Messi?

Are there athletes with very high strength levels that show low rugby performance in your data?

The authors mentioned in your discussion that a 1RM > 1.3 kg*bm-1) might be optimal. Could you expand on this? That is highly relevant for the strength and conditioning specialist. How valid is that data?

Methodological concerns:

Why have you chosen the same 20-40-60-80kg progression for bench press as well as a for the squat exercise? That is in many cases below than 50% (squats) of the 1RM.

Can you explain please how the data of the power testing was selected (best rep, or the average of three reps)?

Table two: Do you present individualized data on the power level in relation to the body weight?

Discussion

In the discussion, the authors have drawn some beautiful interrelations, like the morphological part. I would love to learn more about possible outliers and see if they can help to produce additional relevant hypothesis.

All the best

Author Response

We thank the three reviewers for their overwhelmingly positive appraisal of our paper and for the opportunity to resubmit for their reconsideration. We feel that the manuscript is now much improved and hope it meets with the reviewers’ approval. The comments have been amended as suggested (detail in red within the manuscript) within the paper.

The research group investigated an important aspect of physical prerequisites and playing level. However, in my view, the goal of a case study, better pilot observation, is to find better hypotheses. So, the authors made their conclusion to early based on correlation instead of causational modeling.

 If we take into account that the professional players have significant more training history it is no surprise that they have shown a higher maximum strength and peak power levels compared to academy and scholarships levels. However, I have some, conceptional methodological and causational clues for the authors.

 I would recommend starting to frame the article with a rugby performance model and trying to integrate these findings in this model. Are there athletes with weak strength levels, which also show high rugby performance in your data or in practice like in football Messi? Are there athletes with very high strength levels that show low rugby performance in your data?

-       We appreciate the reviewer’s comments. However, the aim of the study was to compare load-velocity and load-power relationships across different playing standard within the same club. Previous work has compared power profiles in athletes of different playing standards but at different clubs, thus providing room for our work. It is beyond the scope of our study to provide causational modelling. Regarding the athletes that display high and low performance, rugby league is dynamic in nature and requires the performance of a plethora of different skills and abilities. Thus, it is not plausible (for the purpose of our study) to quantify match ‘performance’ per se, beyond categorising athletes into playing standards. Naturally, there were some ‘weaker’ and ‘stronger’ athletes across the playing standards (these would be illustrated within the standard deviations). However, it would be too reductionist to analyse these athletes.

 The authors mentioned in your discussion that a 1RM > 1.3 kg*bm-1) might be optimal. Could you expand on this? That is highly relevant for the strength and conditioning specialist. How valid is that data?

-       Bench press 1RM was correlated to optimal power in the scholarship group (relative strength = 1.03 ± 0.12 kg.bm) but not in the 1st grade or academy players (relative strength = 1.43 ± 0.14 and 1.30 ± 0.15 kg.bm, respectively). Thus, we were suggesting that once an athlete has a relative strength level of > 1.30 kg.bm, like the 1st grade and academy players, then other qualities (e.g. velocity or rate of force development) should be focussed on when aiming to enhance optimal power. That is, the scholarship players should focus on strength training until they attain a relative strength of >1.30 kg.bm, after this point they should aim to increase maximal velocity. We have made some amendments in text to improve the clarity ‘These data suggest that once players are relatively strong enough (i.e. a 1RM of > 1.3 kg·bm-1, like the 1st grade and academy players) then other physical attributes must be focused upon.’

 Methodological concerns:

 Why have you chosen the same 20-40-60-80kg progression for bench press as well as a for the squat exercise? That is in many cases below than 50% (squats) of the 1RM.

-       During rugby league match play, athletes are required to produce velocity and strength efforts against absolute conditions, independent of their relative strength (i.e. percentage 1RM). Moreover, previous work indicates that strength and power adaptations are underpinned by the ability to overcome absolute loading conditions (Baker & Nance, 1999; Hakkinen & Komi, 1985; Mayhew et al., 1997). Such a loading strategy has been deemed effective in determining differences in power between populations (see references 3-7 in the manuscript). A brief rationale has been included in the Methods section; ‘We opted to use absolute loading conditions, rather than relative, as this better reflects match demands i.e. players are required to express velocity and power against absolute loads, irrespective of their individual strength. Such an approach has been adopted previously [3-7].’

 Can you explain please how the data of the power testing was selected (best rep, or the average of three reps)?

-       The average of three repetitions was selected for analysis. This has now been included in the manuscript ‘The average of three repetitions was selected for analysis.’

Table two: Do you present individualized data on the power level in relation to the body weight?

-       The data presented in Table 2 is the absolute power values as opposite to the relative (per body mass) power. We opted to use this as bench press and squat are not ballistic thus the body mass is not being propelled.

 Discussion

In the discussion, the authors have drawn some beautiful interrelations, like the morphological part. I would love to learn more about possible outliers and see if they can help to produce additional relevant hypothesis.

 -       We thank the review for this comment. As mentioned above, it is likely that there are outliers in the data illustrated by ‘weaker’ and ‘stronger’ athletes in different playing standards. The standard deviations presented would provide some indication of this. However, it would be too reductionist to analyse these, especially given the aim of the study. Moreover, we do not have the morphological data to explain these differences and were only alluding to possible mechanisms that have been investigated previously.